# Monitoring peripheral perfusion in sepsis associated acute kidney injury: Analysis of mortality

Ana Carolina de Miranda[1]*, Igor Alexandre Cortês de Menezes[1], Hipolito Carraro Junior[2‡], Alain Márcio Luy[2‡], Marcelo Mazza do Nascimento[1]

1 Department of Internal Medicine, Hospital de Clínicas, Federal University of Paraná, Curitiba, Paraná, Brazil, 2 Intensive Care Unit, Hospital de Clínicas, Federal University of Paraná, Curitiba, Paraná, Brazil

☯ These authors contributed equally to this work.
‡ These authors also contributed equally to this work.
* miranda.anacarolina@gmail.com

**Data Availability Statement:** All figures and Supporting Information files are available from the Figshare database (doi: 10.6084/m9.figshare.12918164).

## Abstract

Microcirculatory disorders have been consistently linked to the pathophysiology of sepsis. One of the major organs affected is the kidneys, resulting in sepsis-associated acute kidney injury (SA-AKI) that correlates considerably with mortality. However, the potential role of clinical assessment of peripheral perfusion as a possible tool for SA-AKI management has not been established. To address this gap, the purpose of this study was to investigate the prevalence of peripheral hypoperfusion in SA-AKI, its association with mortality, and fluid balance. This observational cohort study enrolled consecutive septic patients in the Intensive Care Unit. After fluid resuscitation, peripheral perfusion was evaluated using the capillary filling time (CRT) and peripheral perfusion index (PI) techniques. The AKI was defined based on both serum creatinine and urine output criteria. One hundred and forty-one patients were included, 28 (19%) in the non-SA-AKI group, and 113 (81%) in the SA-AKI group. The study revealed higher peripheral hypoperfusion rates in the SA-AKI group using the CRT (OR 3.6; 95% CI 1.35–9.55; p < 0.05). However, this result lost significance after multivariate adjustment. Perfusion abnormalities in the SA-AKI group diagnosed by both CRT (RR 1.96; 95% CI 1.25–3.08) and PI (RR 1.98; 95% CI 1.37–2.86) methods were associated to higher rates of 28-day mortality (p < 0.01). The PI's temporal analysis showed a high predictive value for death over the first 72 h (p < 0.01). A weak correlation between PI values and the fluid balance was found over the first 24 h (r = - 0.20; p < 0.05). In conclusion, peripheral perfusion was not different intrinsically between patients with or without SA-AKI. The presence of peripheral hypoperfusion in the SA-AKI group has appeared to be a prognostic marker for mortality. This evaluation maintained its predictive value over the first 72 hours. The fluid balance possibly negatively influences peripheral perfusion in the SA-AKI.

**Funding:** The authors received no specific funding for this work.

**Competing interests:** The authors have declared that no competing interests exist.

## Introduction

Sepsis consists of a dysregulated host response secondary to an infection, which leads to organ dysfunction [1]. Even though clinical outcomes have improved [2], sepsis still constitutes a large proportion of the critically ill population and remains one of the most significant causes of death globally [3]. Furthermore, it is the most contributing determinant for developing acute kidney injury (AKI) [4], accounting for 50% of cases in the intensive care unit [5], thus increasing morbidity and mortality [6].

The crucial steps for sepsis management are fast identification, early antibiotic therapy, fluid administration, and commonly vasoactive drugs [7]. Recently, tissue perfusion monitoring has become an essential tool for sepsis recognition [8], analysis of its therapeutic response [9], and prognosis [10]. Although the usual hemodynamic targets are based on the macro-circulation goals, robust evidence has shown that persistent microcirculatory blood flow disturbances occur despite the restoration of macro-hemodynamics and are associated with organ failure and lower rates of survival [10, 11]. This dissociation between macro and micro-hemodynamics demonstrates the importance of accessing the perfusion disturbances and septic patients' microcirculation.

The skin is one of the most accessible organs in the human body [12], which allows clinicians to evaluate peripheral perfusion of cutaneous microvessels with noninvasive bedside parameters such as the capillary refill time (CRT) and the peripheral perfusion index (PI). The rationale for peripheral perfusion monitoring is based on the concept that peripheral tissues, such as skin and muscles, are the first to be affected by impaired blood flow in sepsis and the last to be reperfused after resuscitation [8, 13].

Robust research has shown that noninvasive peripheral perfusion can potentially predict outcomes [14], and be used as a reasonable goal for resuscitation in non-selected septic patients [9]. However, the literature is scarce regarding the importance of this type of monitoring for each kind of organ dysfunction associated with sepsis since the various dysfunctions imply prognostic heterogeneity [1, 3]. In this context, AKI deserves special attention because although it has an incomplete understanding of its pathophysiology, it seems to involve microcirculatory dysfunction, besides the inflammation and metabolic reprogramming [4]. Another possible mechanism that could link AKI and peripheral microcirculatory hypoperfusion in sepsis would be the persistent positive fluid balance [15], which has not been previously evaluated.

Although the peripheral perfusion has been extensively studied in non-selected septic patients, to the best of our knowledge, the potential role of clinical assessment of peripheral perfusion as a possible tool for SA-AKI management has not been established. To achieve this goal, this present observational study was designed to investigate the prevalence of peripheral hypoperfusion using CRT and PI between SA-AKI and non-SA-AKI groups, its association with 28 days in-hospital mortality, and correlation with positive fluid balance.

## Materials and methods

### Study design, setting & participants

This observational cohort study was performed in the 22-bed of two intensive care units (ICU) within the Hospital de Clínicas, Federal University of Paraná, Brazil, between February 2019 and December 2019. All participants or their legal representatives provided written informed consent, and the "Human Research Ethics Committee" of the Hospital de Clínicas, Federal University of Paraná, approved the research (protocol: 3.142.086/2019).

Consecutive adult patients (aged ≥18 years) with sepsis admitted to the ICU or within 24 hours after sepsis onset in patients previously admitted for other causes were considered

eligible. The exclusion criteria used in the study to minimize potential confounding factors were pregnancy, inaccessible peripheral perfusion (Severe Hypothermia, Raynaud's Syndrome, Peripheral Arterial Occlusive Diseases, and Scleroderma), Kidney Chronic Disease stage 5, and no informed consent.

After selection, patients were distributed in 2 groups based on the diagnosis or not of AKI.

**Clinical definitions.**   According to the most recent Sepsis consensus, this syndrome is defined as the presence of an infection combined with an acute change in the "Sequential Organ Failure Assessment" (SOFA) score of two points or more [1]. Septic shock remains a subset of sepsis cases wherein, despite adequate resuscitation, patients have hypotension requiring vasopressors to maintain mean arterial blood pressure (MAP) above 65 mmHg and have elevated serum lactate concentration $\geq 2$ mmol/L [1].

AKI was defined according to Kidney Disease Improving Global Outcomes (KDIGO), which included urine output < 0.5 ml/kg/h for 6 hours and/or increase of serum creatinine by $\geq 0.3$ mg/dl (x 26.5 μmol/l) within 48 h or increased serum creatinine to $\geq 1.5$ times baseline which is known or presumed to have occurred within the previous seven days [16]. Baseline kidney function was defined as the most recent outpatient, non-emergency department serum creatinine concentration between 7 and 365 days before sepsis diagnosis [17].

## Study protocol

All selected patients were treated according to our local institution recommendations, which has been adapted from "Surviving Sepsis Campaign" (SSC) guidelines: 30 ml/kg of ringer lactate or crystalloid fluid were administrated at the discretion of the patients' physicians over the 1 hour of sepsis diagnosis which was continues until lack of response to passive-leg raising (the cutoff value to discriminate fluid responders was an increase in cardiac output of 13%) or no respiratory variations of inferior vena cava diameter (the cutoff was 18%) and if MAP remained < 65 mmHg, norepinephrine or a combination of vasopressors were used to maintain MAP $\geq 65$ mmHg [7]. The hemodynamic targets were as follows: MAP > 65 mmHg, urine output > 0.5 ml/kg/h and central venous oxygen saturation (ScvO$_2$) > 70%. Intensivists were blinded for peripheral perfusion variables to avoid possible treatment interferences. The patients were followed until 28 days of sepsis diagnosis or discharge from the hospital.

Information collected included demographic characteristics, admission diagnosis, the focus of infection and comorbidities, "Acute Physiology and Chronic Health Evolution II" (APACHE II), and SOFA scores. Assessment of patients occurred within 24 h after admission in ICU with the diagnosis of sepsis or within 24 h after the onset of sepsis in patients previously admitted for other causes. Hemodynamic parameters (if available), metabolic, and peripheral variables were measured after fluid resuscitation between 6 and 24 hours of sepsis diagnosis. Cumulative fluid balance was calculated as the fluid input (enteral and parenteral) minus fluid output (urine output, fluid from drains, and gastric aspiration) recorded every 24 hours [18]. Serum creatinine and urine output were observed for seven days after sepsis diagnosis.

**Peripheral perfusion assessment.**   It was evaluated using a combination of the CRT and the PI.

CRT was defined as the time required for the distal capillary bed of the index finger to recover its color after pressure has been applied for 10 seconds to cause blanching. The time to return of standard color was recorded with a mobile phone chronometer. A delayed return of standard color $\geq 3$ s was considered abnormal [9, 19]. CRT was assessed after fluid resuscitation in the first 24 hours.

The PI is a method derived from the photoelectric plethysmography signal of a pulse oximeter, which provides a noninvasive indicator of peripheral vasomotor tone and peripheral

perfusion. It was measured after fluid resuscitation by attaching a pulse oximeter probe (ELERA, Hubei, China). After signal stabilization, PI was recorded every 30 seconds for 5 minutes, and the average of the values was calculated and used as a reference value. The PI analyses were made on the first, second, and third days of sepsis diagnosis. A PI < 1.4 was the cutoff point for determining abnormal peripheral perfusion [20].

In order to avoid "observer bias," the clinical assessment of peripheral perfusion was conducted by only one trained researcher who was not involved in patient care. The ambient bedside temperature was controlled at 22˚C. It was made in a supine decubitus position and performed in the upper limb without an intraarterial catheter for MAP measurement.

To speculate if the fluid balance could be a link factor that associates peripheral hypoperfusion with the development and prognosis of patients with SA-AKI, a correlation between the 24 h cumulative fluid balance and PI values 24 h after sepsis diagnosis.

**Outcomes.** The primary outcomes were the prevalence of peripheral hypoperfusion between SA-AKI and non-SA-AKI groups and the in-hospital mortality rates observed within 28 days, associated with peripheral hypoperfusion in the SA-AKI group. The secondary outcomes included the correlation between fluid balance and PI values in the SA-AKI group; the prognostic significance of serial changes in PI values over the first 72 hours in the SA-AKI group; and multivariate analysis of peripheral perfusion parameters as independent predictors.

## Analytical approach

The Shapiro-Wilk test was used to assess the normality of the sample. Parametric data were described through mean ± standard deviation, while for non-parametric data, the median and interquartile range was used. Percentages represented proportions. First, we divided the patients into SA-AKI and non-SA-AKI groups, second into survivors and nonsurvivors of the SA-AKI group, and tested differences between groups means by Student's tests or Mann-Whitney U test if not normally distributed. Comparative analyses of the prevalence of peripheral hypoperfusion between SA-AKI and non-SA-AKI groups and the SA-AKI mortality rates between groups with predefined cutoffs for abnormal perfusion (CRT $\geq$ 3 seconds and PI < 1.4) were performed using Fisher's exact test. For the serial evaluation of peripheral perfusion among SA-AKI patients first, a logarithmic transformation of the PI data (continuous variable) was needed. Thus, The Linear Model Mixed Effects test was performed. Thus, a linear-mixed effects model with random intercept and slope was used to test hypotheses related to the group (survivors and nonsurvivors), time (first, second and third days), and interaction between time and group. Group was considered a fixed effect, and the patient a random effect. The Bonferroni test was subsequently added for correction due to multiple comparisons. A logistic regression model was used for the multivariate analysis of peripheral perfusion parameters as independent predictors. Correlation tests between continuous variables were performed using the Spearman test. A p-value < 0.05 was considered as statistically significant. All reported p-values are two-sided. The statistical programs IBM SPSS Statistics and GraphPad Prism 6 were used for all analyses.

The sample size was calculated based on previous local and pilot studies in sepsis [21]. Assuming 28-day mortality around 45–50% in the SA-AKI group and a 40% prevalence of altered peripheral perfusion, estimating a relative risk ratio of 2, we determined that the enrollment of 113 patients in the SA-AKI group would provide a 90% power to detect a mortality difference at an alpha level of 0.05. The previous studies also demonstrated that local non-SA-AKI patients represented approximately 15–20% of septic patients. Considering a peripheral hypoperfusion prevalence of 8% in patients without AKI, 28 patients in the control group were enrolled to have 90% power to reach an odds ratio of 7.0 between groups.

This study followed STROBE guidelines for reporting results.

## Results

During the study period, 141 septic patients were included after fluid resuscitation and had subsequent peripheral perfusion assessment (Fig 1). The clinical-demographic and hemodynamic data of these patients are listed in the S1 Table. Taken as a whole, these data describe a heterogeneous critically ill population, classical finding of septic patients. The SA-AKI group had higher APACHE and SOFA scores, lower urine output, needed more vasoactive drugs, and higher arterial lactate levels than no SA-AKI group. There were no differences in the clinical data, infection source, confirmed culture, C-reactive protein, Procalcitonin, and other hemodynamic parameters between groups.

Fig 2 shows the prevalence of peripheral hypoperfusion between SA-AKI and non-SA-AKI groups. There was no significant (ns) difference between groups using PI measurements (Fig 2A). The bivariate analysis revealed a statistically relevant difference between groups when analyzed by the CRT method (OR 3.6; 95% CI 1.35–9.55), being higher in the SA-AKI group (Fig 2B). However, after adjustment for the use of vasoactive drugs in the multivariate analysis, the association lost its statistical significance (OR = 2.6; 95% CI 0.93–7.33; p = 0.07).

S3 Table shows a prevalence of peripheral hypoperfusion subanalyses based on the time of AKI diagnosis. There was no difference between patients who had AKI diagnosis over the first 24 hours and Patients with AKI diagnosis after 24 hours (S3C).

The 28-day in-hospital mortality of the SA-AKI group was 55% (62/113), which was higher than the non- SA-AKI group 25% (7/28) (p < 0.01). The comparison of the clinical-demographic and hemodynamic between survivors and nonsurvivors are listed in the S2 Table. The nonsurvivor group had higher APACHE, and SOFA scores needed higher noradrenaline doses, more often used of vasopressin, and had higher heart rate after fluid resuscitation than survivors. There were no differences in the clinical data, infection source, confirmed culture, C-reactive protein, Procalcitonin, need of hemodialysis, and other hemodynamic parameters between survivors and the nonsurvivors.

On the analysis of outcomes (Fig 3A and 3B), abnormalities in peripheral perfusion in the SA-AKI group evaluated by using both CRT (RR 1.96; 95% CI 1.25–3.08) and PI (RR 1.98; 95% CI 1.37–2.86) techniques were associated statistically relevant higher rates of 28-day in-hospital mortality. In multivariable analyses (Tables 1 and 2), higher SOFA score and altered peripheral perfusion by both methods were independently associated with an increased adjusted rate of 28-day in-hospital mortality.

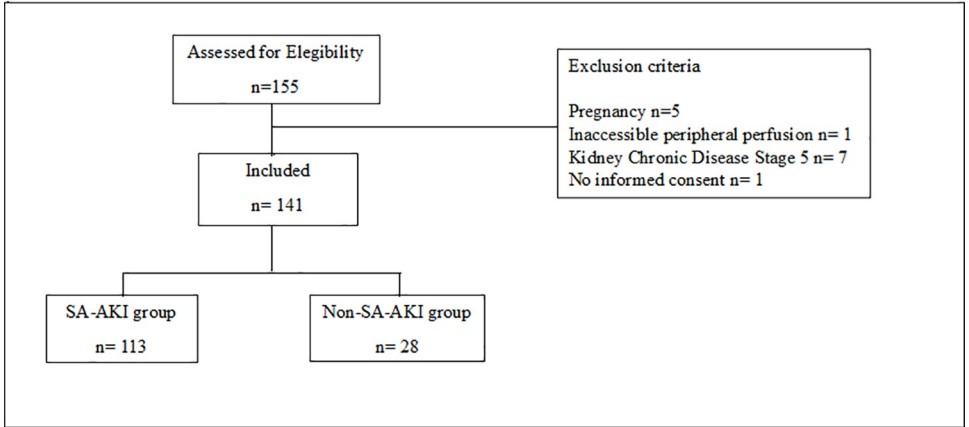

**Fig 1. Flow-chard of study.**

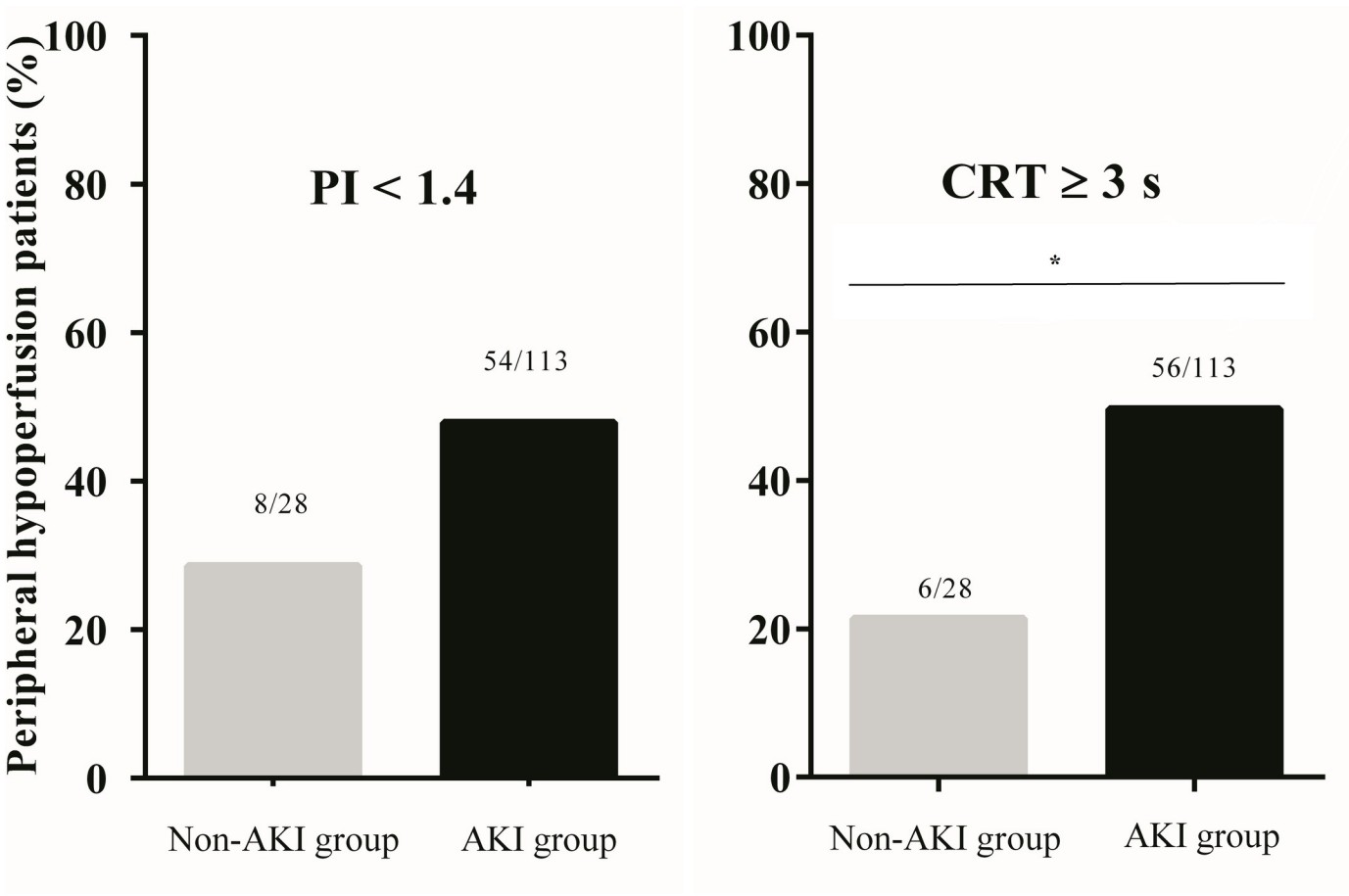

**Fig 2. Prevalence of peripheral hypoperfusion after fluid resuscitation between SA-AKI and non-SA-AKI groups.** (A) Peripheral Perfusion Index ns (B) Capillary Refill Time *p < 0.05.

As shown in Fig 4, we performed a PI's serial assessment during the first three days after fluid resuscitation in the SA-AKI group. There was no interaction between days and groups (p = 0.43) (Fig 4A). There was no significant alteration of PI values for both groups over time (p = 0.33 for intragroup analyses) (Fig 4B). As illustrated below, there was a significant difference between survivors and nonsurvivors throughout the entire evaluated period (Fig 4C).

Fig 5 shows a weak but statistically significant negative correlation between PI values and fluid balance within the first 24 hours of sepsis diagnosis in the SA-AKI group.

## Discussion

There is an essential role of microcirculatory disorders in the origin and progression of sepsis-related organic dysfunctions [22, 23]. Kidney involvement in sepsis is frequent [24], resulting in sepsis-associated acute kidney injury and unacceptable morbidity and mortality [25, 26]. The results of this study demonstrated that SA-AKI patients showed two times higher mortality rates than the non-SA-AKI group and therefore corroborated the findings from the literature. Moreover, regarding the diagnostic/therapeutic role of the clinical monitoring of peripheral perfusion in SA-AKI, this study brings several new pieces of evidence that can contribute to support its widespread clinical use by nephrologists and non-nephrologists.

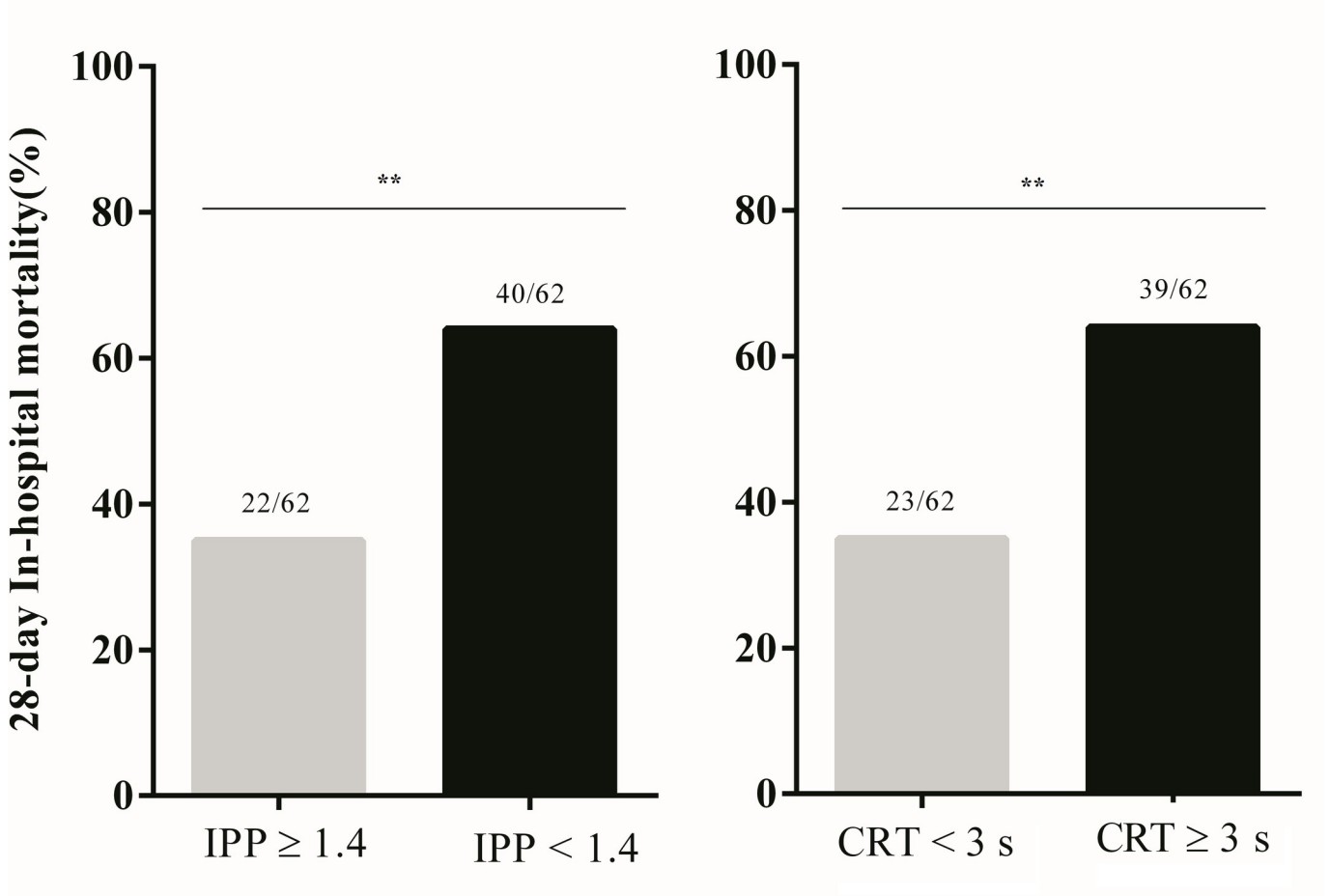

**Fig 3. The 28-day in-hospital mortality rates associated with peripheral hypoperfusion in the SA-AKI group.** (A) Peripheral perfusion index $^{**}$p < 0.01; (B) Capillary Refill Time $^{**}$p < 0.01.

Microcirculatory blood flow disturbances are unequivocally recognized as independent prognostic markers in sepsis patients [27, 28], and preclinical studies have shown that it appears to play an essential role in the pathophysiology of SA-AKI [29, 30]. However, to the best of our knowledge, there is no clear evidence demonstrating a possible link between microvascular disorders, measurable by bedside methods by caregivers, and the development of

**Table 1. Prognostic analyses of mortality–peripheral perfusion index.**

| Bivariate | | | | Multivariate | | |
|---|---|---|---|---|---|---|
| | Odds Ratio | CI 95% | P-value | Odds Ratio | IC 95% | P-value |
| PI < 1.4 | 4.81 | 2.15–10.75 | 0.00** | 2.97 | 1.24–7.13 | 0.01* |
| SOFA score | 1.21 | 1.08–1.36 | 0.00** | 1.12 | 0.98–1.28 | 0.10 |
| Urea nitrogen serum | 1.01 | 1.00–1.02 | 0.02* | 1.01 | 0.99–1.02 | 0.09 |
| Lactate | 1.40 | 1.05–1.87 | 0.02* | 1.15 | 0.83–1.58 | 0.41 |
| Heart Rate | 1.02 | 1.00–1.03 | 0.03* | 1.01 | 0.99–1.03 | 0.28 |

$^{*}$p < 0.05

$^{**}$p < 0.01.

**Table 2. Prognostic analyses of mortality–capillary refill time.**

| Bivariate | | | | Multivariate | | |
|---|---|---|---|---|---|---|
| | Odds Ratio | CI 95% | P-value | Odds Ratio | IC 95% | P-value |
| CRT $\geq$ 3 s | 3.39 | 1.55–7.37 | 0.00** | 2.58 | 1.08–6.18 | 0.03* |
| SOFA score | 1.21 | 1.08–1.36 | 0.00** | 1.21 | 1.05–1.40 | 0.00** |
| Urea nitrogen serum | 1.01 | 1.00–1.02 | 0.02* | 1.02 | 0.99–1.04 | 0.10 |
| Lactate | 1.40 | 1.05–1.87 | 0.02* | 1.21 | 0.88–1.65 | 0.23 |
| Heart Rate | 1.02 | 1.00–1.03 | 0.03* | 0.74 | 0.22–2.44 | 0.62 |

*p < 0.05

**p < 0.01.

SA-AKI. To test the hypothesis, this study was undertaken to investigate if SA-AKI patients could present a higher prevalence of peripheral hypoperfusion than the non-SA-AKI group. Interestingly, our results showed significant differences in perfusion between groups using at least one method (CRT), being worse in the SA-AKI group.

The sympathetic stimulus in reducing perfusion at the fingertips is well known [31], and one could argue the possible influence effects of catecholamines on the CRT values. Thus, the use of vasoactive drugs analysis was used to adjust for possible differences among groups. After multivariate adjustment, the difference in CRT between the groups lost its statistical significance, suggesting that the use of noradrenaline as a vasopressor seems to interfere in the existence of peripheral hypoperfusion in SA-AKI patients. Hence, the CRT measurement was considered a confounding variable, and the peripheral perfusion seems not to be determinant on the prevalence of SA-AKI, at least, after hemodynamic resuscitation, although preclinical studies robustly demonstrate the association between microcirculatory disorders and SA-AKI development [29, 30]. A possible explanation for this lack of association is the differences between renal and skin microcirculatory structures [32–34]. Such disagreements are also

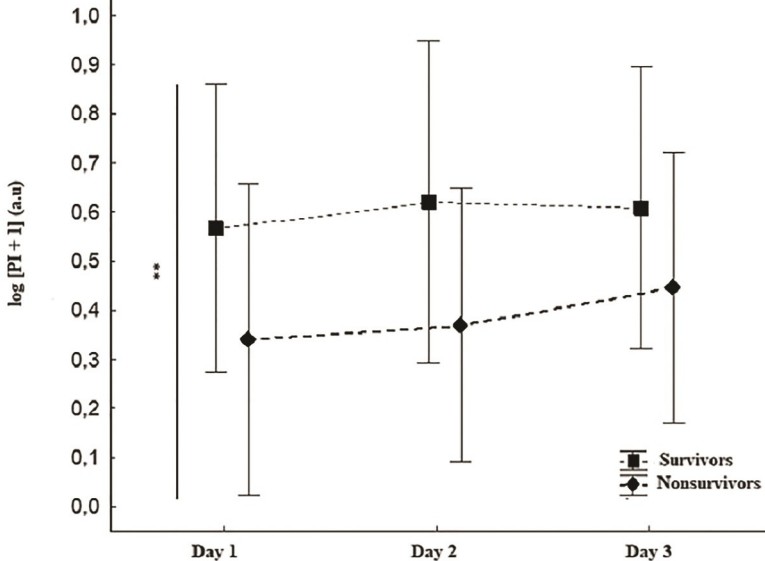

**Fig 4. Prognostic significance of PI's values over the first 72 hours in the SA-AKI group.** (A) Group and time interaction analyses ns (B) Intragroup analyses ns (C) Intergroups analyses (survivors and nonsurvivors) **p < 0.01.

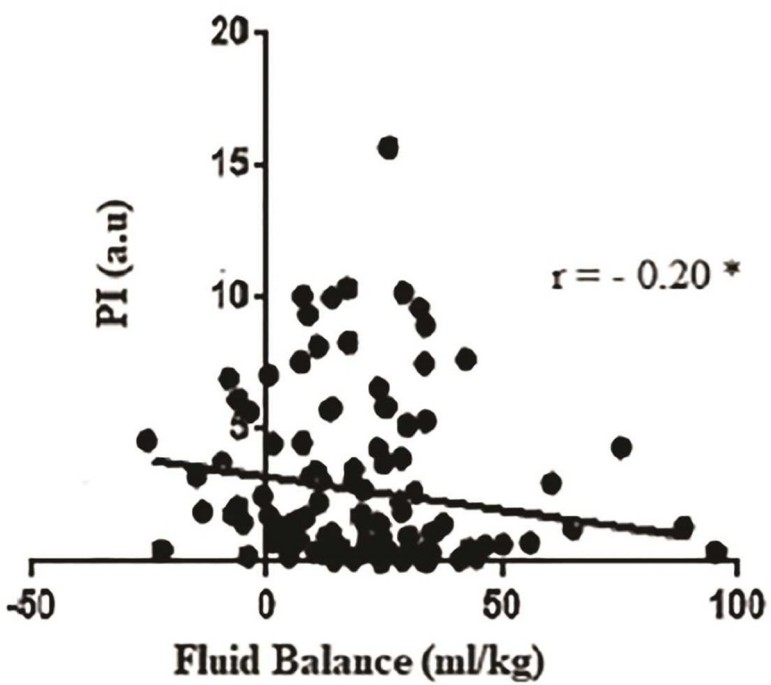

**Fig 5. Correlation between fluid balance and PI's values within 24 h of sepsis diagnosis in the SA-AKI group.**
*p < 0.05.

demonstrated by unequal vasoactive mediators that act on the local homeostasis [32, 34], which would mean that skin perfusion might not necessarily mirror renal perfusion. Curiously, contradicting our results, a study by Brunauer et al. (2016) showed positive correlations between skin perfusion and visceral organ vascular tone, including the kidney, even though the anatomic-physiological differences cited [35].

It could also be argued that dynamic circulatory changes attributed to sepsis lead to the different time-related clinical significance of peripheral perfusion values [36]. Early stages of sepsis are characterized by a hemodynamic coherence between macro and microcirculation. Thus, over the first hours of sepsis diagnosis, systemic hemodynamic variable correction improves tissue perfusion, and oxygenation witch implies normal peripheral perfusion [37]. However, despite the correction of macro hemodynamic parameters and high cardiac index, some patients evaluate impaired microcirculation resulting in organ dysfunction that correlates considerably with mortality [10, 11, 19, 28]. Thus, a possible temporal interference could explain the absence of peripheral perfusion differences between groups. However, to reduce the comparative bias, our study selected the same assessment time in all groups (after fluid resuscitation between 6 and 24 hours of sepsis diagnosis), which does not explain the findings.

Finally, it is also worth noting that the peripheral perfusion was evaluated only in a "single-measurement." Thus, this study's design could not be used to estimate whether those patients with peripheral hypoperfusion were at risk for later development of AKI. In fact, a very recent study reinforces this hypothesis. Wiersema et al. (2020) demonstrated that a prolonged CRT, among other clinical findings, was able to predict AKI development in critically ill patients [38]. Therefore, new studies are necessary to establish the predictive value of peripheral perfusion assessment for AKI development in septic patients.

Several robust human studies have been shown that the predictive value of persistent peripheral hypoperfusion occurs regardless of the correction of systemic hemodynamic [36].

Indeed, recent studies of He et al., [39] Rasmy I et al., [31] Bourcier et al., [40] and Lara et al., [14] who used the same methods of this study (CRT or PI) in non-selected septic patients have also demonstrated this ability. Now, this study showed specifically in the SA-AKI scenario that peripheral hypoperfusion evaluated by both techniques was an early and independent prognostic marker for mortality. Additionally, the literature had not yet provided clear data on how many days, during the patient assessment, peripheral perfusion monitoring could be used to discriminate the prognosis. Thus, one of the main findings of this study was that unfavorable evolution of the peripheral perfusion in the SA-AKI, following fluid resuscitation, maintained its predictive value, at least, over the first 72 hours of starting the syndrome management. In this evaluation, it is worth noting that only PI was included in this protocol since it was a continuous variable, different from CRT, which was considered in this study only as a dichotomous variable. Thus, further studies should be carried out to assess whether the CRT also has predictive value, like PI in a serial evaluation, using an ordinal or continuous variable.

Interestingly, Sakr et al. performed a serial evaluation of sublingual vessels in septic shock and observed that the differences between survivors and nonsurvivors occurred only after the assessment's 2nd day [10]. One possible explanation for such divergence is that the techniques used in the studies for perfusion assessment have clear operational differences and evaluate different microcirculation levels and differences in the tissues themselves [8]. Hence, there are different vasoactive mediators between sublingual and cutaneous tissues [34, 41], resulting in various degrees of impairment, and accordingly determine disparities in regional blood flow between tissues [42]. In fact, a critical study found no significant correlation between skin and sublingual perfusion in patients with sepsis, corroborating this hypothesis [43]. Thus, it is plausible to admit that the microvascular damage in different tissues may not have the same pathophysiological meaning despite its similarity from a clinical point of view (predictive value for estimating mortality) [36].

Despite the clinical evidence of fluid resuscitation as one of the significant pillars in sepsis management [7], recent studies have shown that excessive and persistent fluid replacement leads to increased organ dysfunction and mortality risk [44, 45]. Several factors as ventilatory disorders, increased intra-abdominal pressure, exacerbation of coagulopathies, cerebral edema, cardiac overload, and acute renal injury exemplify the harmful effects of inappropriate fluid accumulations [15]. Moreover, the fluid overload established by either excessive replacement or reduction of losses is mirrored by a positive fluid balance, usually regulated in human kidneys [46]. On the other hand, the positive fluid balance determines venous congestion, which impairs the capillary pressure gradient and, therefore, the microcirculatory blood flow, suggesting a vicious cycle of organ damage [47]. This hypothesis was proposed in sepsis in a previous study in which patients with high central venous pressure (CVP) presented worse microcirculatory sublingual perfusion [48]. Thus it would be possible to speculate that the fluid balance could be a link factor that associates peripheral hypoperfusion with the development and prognosis of patients with SA-AKI. However, until now, this hypothesis had not been verified. Our results found a weak but significant negative correlation between the fluid balance and PI values among SA-AKI patients over the first 24 hours of sepsis diagnosis. This result strongly suggests possible participation, at least partial, of venous congestion, secondary to positive fluid balance, in peripheral hypoperfusion genesis. However, this study did not verify any direct measurement of venous pressure of patients. Therefore, new studies should be performed to confirm this proposition.

The current study has some limitations. First, our observations were performed in only one center. Thus, a multicenter study is necessary to confirm the findings. Secondly, our data were limited to a short time frame in the first 24 hours regarding the correlation between fluid

balance and peripheral perfusion. Therefore, limit conclusions related to the temporal evolution of this association. Third, the proposed design was non-interventional. Thus, the association between peripheral hypoperfusion among SA-AKI patients and mortality does not prove its causal relationship, although the literature suggests this hypothesis. Fourthly, although our sample was performed consecutively, it was mostly represented by septic shock, limiting our findings to mild cases of the syndrome or other settings outside the ICUs. Finally, this study evaluated patients consecutively to avoid any treatment bias. Thus, the groups inevitably did not have the same number of patients. An "underpowered sample" cannot be ruled out as a cause of the negative result regarding hypoperfusion and the presence/absence of SA-AKI. However, if this difference really occurs, our results suggest that it has lower clinical significance.

In conclusion, peripheral hypoperfusion differences were found among patients who evolved or not with Acute Kidney Injury Associated with Sepsis. However, this had probably occurred due to the use of vasoactive drugs. The presence of these peripheral perfusion abnormalities demonstrates to be significant prognostic markers for mortality in SA-AKI patients. Besides, this evaluation maintained its predictive value over the first 72 hours. The fluid balance possibly negatively influences peripheral perfusion in the SA-AKI. Finally, our results add to the literature support the use of peripheral perfusion as a prognostic evaluation in SA-AKI patients.

## Supporting information

**S1 Table. The demographic, clinical, and hemodynamic of septic patients after fluid resuscitation.**
(PDF)

**S2 Table. The demographic, clinical and hemodynamic of SA-AKI group after fluid resuscitation.**
(PDF)

**S3 Table. Prevalence of peripheral hypoperfusion in first 24 hours between groups.**
(PDF)

**S1 File. Database.** All Patients.
(SAV)

**S2 File. Database.** SA-AKI Patients.
(ZSAV)

## Acknowledgments

The authors would like to thank Prof. Dr. Marcia Olandoski for the technical assistance in statistical analysis, and the Department of Internal Medicine (Hospital de Clínicas, Federal University of Paraná) for the facilities used to perform this study and precious learning support.

## Author Contributions

**Conceptualization:** Ana Carolina de Miranda, Igor Alexandre Cortês de Menezes, Marcelo Mazza do Nascimento.

**Data curation:** Ana Carolina de Miranda.

**Formal analysis:** Ana Carolina de Miranda.

**Investigation:** Ana Carolina de Miranda, Marcelo Mazza do Nascimento.

**Project administration:** Igor Alexandre Cortês de Menezes.

**Supervision:** Igor Alexandre Cortês de Menezes, Marcelo Mazza do Nascimento.

**Visualization:** Ana Carolina de Miranda.

**Writing – original draft:** Ana Carolina de Miranda.

**Writing – review & editing:** Igor Alexandre Cortês de Menezes, Hipolito Carraro Junior, Alain Márcio Luy, Marcelo Mazza do Nascimento.

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
