## [Decision Letter · Decision Letter 0]

20 Aug 2020

PONE-D-20-24607

Monitoring Peripheral Perfusion in Sepsis associated Acute Kidney Injury: Analysis of Clinical Outcomes

PLOS ONE

Dear Dr. de Miranda,

Thank you for submitting your manuscript to PLOS ONE. After careful consideration, we feel that it has merit but does not fully meet PLOS ONE’s publication criteria as it currently stands. Therefore, we invite you to submit a revised version of the manuscript that addresses the points raised during the review process.

We look forward to receiving your revised manuscript.

Kind regards,

Corstiaan den Uil

Academic Editor

PLOS ONE

Journal Requirements:

2.Thank you for stating the following financial disclosure:

 [The funders had no role in study design, data collection and analysis, decision to publish, or preparation of the manuscript.].

3.Thank you for including your ethics statement: 'All participants or their legal representatives provided written informed consent, and the Research Ethics Committee approved the research of the Federal University of Paraná (protocol: 3.142.086/2019).'  

(a) Please amend your current ethics statement to include the full name of the ethics committee/institutional review board(s) that approved your specific study.  

(b) Once you have amended this/these statement(s) in the Methods section of the manuscript, please add the same text to the “Ethics Statement” field of the submission form (via “Edit Submission”).

Reviewers' comments:

Reviewer's Responses to Questions

**Comments to the Author**

1. Is the manuscript technically sound, and do the data support the conclusions?

Reviewer #1: Partly

Reviewer #2: Yes

2. Has the statistical analysis been performed appropriately and rigorously? 

Reviewer #1: No

Reviewer #2: I Don't Know

3. Have the authors made all data underlying the findings in their manuscript fully available?

Reviewer #1: Yes

Reviewer #2: Yes

4. Is the manuscript presented in an intelligible fashion and written in standard English?

Reviewer #1: Yes

Reviewer #2: Yes

5. Review Comments to the Author

Reviewer #1: The study addressed an interesting clinical question. However, there are some issues need to be addressed.

1. "To the best of our knowledge, this is the first study to comprehensively and accurately address the importance of monitoring peripheral perfusion in Sepsis associated AKI (SA-AKI)."---this statement is not true because peripheral perfusion has been extensively explored. I suggest to rephrase this sentence to a more focused area with novelty.

2. This is a cohort study, not a case-control study; so you need to explicitly declared this.

3. The time window for the PI and CRT measurement should be explicitly declared; and its relatioship to the AKI assessment should also be clarified. This is important for the causal inference.

4. details of the Linear Model Mixed Effects test should be clarified. For example, did you allow slope to differ between individials? or random-effects were just allowed for the intercept term? How did you specify random-effects and fixed effect terms?

5. when you identify independent predictors for AKI, you must clarify the temporal relationship between outcome and predictors. The outcome (AKI) may be present on ICU admission.

6. When you explore the association of fluid balance and PI/CRT, you need to clarify the temporal relationship in the METHOD section.

Reviewer #2: The authors present an interesting manuscript about microcirculation in septic patients with or without acute kidney injury. I have some remarks, split per section, below.

Abstract:

- Could the authors shortly mention here that AKI was defined based on KDIGO on both serum creatinine and urine output?

- Numbers at the beginning of a sentence should be fully spelled

- In the last part, it is mentioned that the presence of peripheral hypoperfusion may be an important prognostic marker. Could the authors add here for what specific outcome this prognostic value is?

Introduction:

- Could the authors elaborate either here or in the discussion about how the ´´phase of sepsis´´ may influence the made measurements? I.e. at certain points patients may have high cardiac index with even shortened CRT before suffering from peripheral hypoperfusion and a prolonged CRT.

- On page 4, there are the numbers 1 and 3 in superscript. Should these be references? Please clarify

- In the last section of the introduction (and in the title), the authors mention clinical outcomes, where later the most focus is on mortality. Could this be specified or changed?

Methods:

- One of the exclusion criteria is inaccessible peripheral perfusion, could the authors provide an example?

- How were missing values handled? I.e. missing urine output data or missing pre-admission creatinine? Was a baseline eGFR assumed?

- The CRT was measured at what place? Index finger? This is important for the reference value.

- How did the authors decide on the limit of 3 secs for the CRT to be abnormal? In the stated reference, 2.4 s is mentioned for septic shock pts and 5 for critically ill, previous papers have stated 4.5 seconds.

- The outcome section of the methods more clearly describes the objective, suggest to write something similar in the last paragraph of the introduction.

- I cannot follow the rationale behind the sample size calculation and the aimed OR of 7.0 between groups, could the authors elaborate?

Results:

- What was the time between admission, study inclusion and fluid resuscitation?

- The authors could consider to either shorten or move table 1 and 2 to the appendices. They are informative but do not provide main answers to the main objectives of the study.

- Did the authors also consider using CRT as a continuous variable?

- Could the authors provide more data on the received fluid resuscitation?

- In figure 4, the difference between PI and changes over time is plotted, splitted for survivors and non-survivors. The title of the figure states the prognostic significance, and a p value. I do not fully comprehend: does this mean that the changes were prognostic (as there was not a significant alteration) or that all measurements, apart from eachother, were consistently different between survivors and non survivors?

Discussion:

- Could the authors specify and clarify what specific pieces of evidence contribute to the support of the use of? CRT? PI? Both? For AKI? Mortality?

- Please change the word humans into something like ´´caregivers´´

- On page 16, the use of CRT dichotomously or continuously is discussed. Perhaps the authors could add a sensitivity analysis using a different cut off value to see whether this changes the main findings? And here discuss the potential consequences?

- In the conclusion, could the authors clarify the sentence : ´´however, the presence of … independently´´. The sentence reads a bit difficult and it is unclear to me what the specific prognostic marker is for (and for what outcome).

6. PLOS authors have the option to publish the peer review history of their article (what does this mean?). If published, this will include your full peer review and any attached files.

Reviewer #1: **Yes: **Zhongheng Zhang

Reviewer #2: No

---

## [Author Response · Author response to Decision Letter 0]

10 Sep 2020

We would like to extend our gratitude to the reviewers for their detailed and instructive comments, and we hope that we have sufficiently addressed the concerns raised. We believe the revised manuscript is much clearer and stronger due to the comments provided. Listed below are changes made in response to reviewers’ comments.

---

## [Editor Report · Decision Letter 1]

14 Sep 2020

Monitoring Peripheral Perfusion in Sepsis associated Acute Kidney Injury: Analysis of Mortality

PONE-D-20-24607R1

Dear Dr. de Miranda,

We’re pleased to inform you that your manuscript has been judged scientifically suitable for publication and will be formally accepted for publication once it meets all outstanding technical requirements.

Kind regards,

Corstiaan den Uil

Academic Editor

PLOS ONE
---

## [Editor Report · Acceptance letter]

17 Sep 2020

PONE-D-20-24607R1 

Monitoring Peripheral Perfusion in Sepsis associated Acute Kidney Injury: Analysis of Mortality 

Dear Dr. de Miranda:

I'm pleased to inform you that your manuscript has been deemed suitable for publication in PLOS ONE. Congratulations! Your manuscript is now with our production department. 

Kind regards, 

on behalf of

Dr. Corstiaan den Uil 

Academic Editor

PLOS ONE